# Chaos Detection by Fast Dynamic Indicators in Reflecting Billiards

**DOI:** 10.3390/e25091251

**Published:** 2023-08-23

**Authors:** Gabriele Gradoni, Giorgio Turchetti, Federico Panichi

**Affiliations:** 1Department of Electrical and Electronic Engineering, School of Mathematical Sciences, University of Nottingham, University Park, Nottingham NG7 2RD, UK; 2Department of Physics and Astronomy, Alma Mater Studiorum, University of Bologna, Viale Berti Pichat 6/2, 40127 Bologna, Italy; 3INdAM Gruppo Nazionale per la Fisica Matematica Piazzale Aldo Moro, 00185 Roma, Italy

**Keywords:** Lyapunov error, reversibility error, Gibbs entropy, wave chaos

## Abstract

The propagation of electromagnetic waves in a closed domain with a reflecting boundary amounts, in the eikonal approximation, to the propagation of rays in a billiard. If the inner medium is uniform, then the symplectic reflection map provides the polygonal rays’ paths. The linear response theory is used to analyze the stability of any trajectory. The Lyapunov and reversibility error invariant indicators provide an estimate of the sensitivity to a small initial random deviation and to a small random deviation at any reflection, respectively. A family of chaotic billiards is considered to test the chaos detection effectiveness of the above indicators.

## 1. Introduction

Electromagnetic cavities exhibit wave chaos that can be predicted by a semi-classical analysis and random matrix theory; see [1] for a two-dimensional open electromagnetic cavity, ref. [2] for closed three-dimensional cavities and [3] for a review. These powerful prediction tools hold true and can be extended to include coupling through antennas and wave-guides, as well as interconnected cavities [4]. Furthermore, the semi-classical treatment of electromagnetic cavities has been tackled extensively in [5,6]. The experimental verification of wave chaos in microwave billiards has been addressed by several international research groups over the last few decades. This extensive work led to the observation of a rich phenomenology originating from complex wave dynamics [7], including: Wave-function scars [8], chaotic dynamics in superconducting billiards [9], time-reversal symmetry breaking [10], nodal domains in rough billiards [11], time-invariance violation at and around exceptional points [12] as well as electromagnetic reverberation [13]. Concerning optical resonators, it has been shown that chaotic dynamics are a key mechanism in asymmetric geometries [14,15,16]. This effect is important in the generation of lasers [17] with directional emissions [18] of eccentric cavities. Chaotic cavities have been employed in a plethora of electromagnetic engineering applications, including time reversal energy focusing [19] and electromagnetic compatibility (EMC) testing [20]. The equations for the propagation of acoustic and electromagnetic waves in resonant cavities are similar and coincide when the medium within the cavity is uniform. Usually, the cavities have a cylindrical or spherical symmetry and the resonant modes can be computed in a closed mathematical form. When the eikonal approximation is applicable, for sufficiently short wavelengths, the symplectic reflection map is integrable [21]. Deformations of the boundary cause the loss of integrability and the emergence of chaotic behavior in the particles or rays trajectories [22]. Even if the dynamics of integrable billiards are still an exciting research topic that continuously unveils novel phenomena, a growing interest has been developing within the scientific community for the properties of chaotic billiards [23,24,25,26,27].

For a sphere, the trajectory of any ray develops in a plane, which is determined by the initial ray position and velocity. The intersection of the sphere with the invariant plane is a circle, where the 2D area preserving the reflection map is integrable.

The symplectic 4D reflection map on a sphere is no longer integrable if the sphere is deformed. Thus, as opposed to a 4D flow, a first integral is no longer available and the 2D Poincaré map cannot be computed, preventing the analysis of the dynamical structures in phase plane portraits. For a cylinder, the trajectory of any ray develops in a plane, which intersects it on an ellipse, where the 2D reflection map is integrable. When the cylinder is deformed, the orbits no longer belong to a plane, the 4D symplectic map is not integrable and the phase portraits of the reflection map cannot be drawn [28]. This is important also in wave propagation within deformed cavities as the evolution of quantum states show footprints of classical trajectories [29].

The Lyapunov error ($E_{n}\left(x\right)$) and the reversibility error ($E_{n}^{R}\left(x\right)$) are dynamic indicators, based on the linear response theory, and allow the testing of the sensitivity of the orbits to small random deviations (see, for instance, [30]). For any fixed number of iterations, this sensitivity can be compared on a set of initial conditions chosen in a phase plane (see, for instance, [31]).

In Section 2, we present a brief introduction of the fast indicator used, their properties and relations. In Appendix B, we provide the definition of Lyapunov and reversibility errors. For a single orbit, the dependence on *n* can be investigated and the limit n→∞, limit of $n^{-1}\;\log E_{n}\left(x\right)$, provides the maximum Lyapunov exponent. $E_{n}^{R}\left(x\right)$ has been shown to be very sensitive to multi-dimensional problems, such as chaos detection in planetary systems (see, for instance, [32]) and in 2D and 3D waveguides (see [33]). In Section 3, we present a numerical analysis of the 2D reflection map in a convex domain, given by a deformed circle. We compare the phase portraits with the color plots of the Lyapunov and reversibility errors computed in a regular grid of phase space.

The mathematical description of the convex billiard and its new parametrization are presented in Appendix A.

To conclude, we consider the transport of particles and rays within the billiard. Given a source of particles or rays within the billiard, the time evolution of the probability density of particles or the energy density of rays is analyzed. Such a density can hardly be determined analytically, even for integrable billiards, and a numerical strategy is presented.

The dynamic indicators could be evaluated at time *t* for the inner points of the billiard after performing an average with respect to the initial ray or particle direction. A detailed analysis of 2D and 3D billiards can be worked out by using the algorithms described here.

## 2. Lyapunov and Reversibility Error Indicators

The dynamic stability of the reflection map can be analyzed with the Lyapunov error indicator (i.e., $E_{n}\left(x\right)$) or with the Reversibility error indicator (i.e., $E_{n}^{R}\left(x\right)$). The former measures the sensitivity of the initial conditions to small random deviations, and the latter measures the sensitivity to small deviations at each reflection. To implement the Reversibility error indicator, we iterate both the randomly perturbed and exact maps’ *n* times, and compute the distance of the final point of the perturbed orbit with respect to the one obtained from the exact orbit. Letting ϵ be the amplitude of the random perturbations, we consider the linear response given by the ϵ→0 limit. Given a symplectic or measure preserving map M(x), defined as a compact manifold of $R^{2d}$, we denote with DM(x) the tangent map where ${\left(DM\right)}_{ij}\left(x\right)=\partial \;M_{i}\left(x\right)/\partial x_{j}$. The orbit $x_{n}$ is obtained by iterating *n* times the map *M*. We introduce the matrix ${𝖫}_{n}$ obtained by taking the products of the tangent map along the orbit: (1)$$\begin{array}{cc} \; & x_{n}=M\left(x_{n-1}\right)\;\;\;\;x_{0}=x\;\;x_{n}=M^{\circ n}\left(x\right)\\ \; & {𝖫}_{n}\left(x\right)=DM\left(x_{n-1}\right)\;{𝖫}_{n-1}\left(x\right)\;{𝖫}_{0}=𝖨\;\;{𝖫}_{n}\left(x\right)=DM^{\circ n}\left(x\right) \end{array}$$
For the initial condition y=x+ϵξ, where ***ξ*** is a random vector with a zero mean and unit covariance matrix $\langle \;\xi \;\xi^{T}\;\rangle =𝖨$, we compute the orbit $y_{n}=M\left(y_{n-1}\right)$ where $y_{0}=y$. We compare the perturbed and the reference orbit by defining the linear response ${𝜩}_{n}$ according to
(2)$$\begin{array}{cc} \; & {𝜩}_{n}\left(x\right)=\lim_{\epsilon \to 0}\;\frac{y_{n}-x_{n}}{\epsilon }=\lim_{\epsilon \to 0}\;\frac{M^{\circ n}\left(x+\epsilon \xi \right)-M^{\circ n}\left(x\right)}{\epsilon }=DM^{\circ n}\left(x\right)\;\xi \equiv {𝖫}_{n}\left(x\right)\;\xi \end{array}$$
The square of $E_{n}\left(x\right)$ is defined as the trace of the covariance matrix $\Sigma_{n}^{2}\left(x\right)$ of the random vector ${𝜩}_{n}\left(x\right)$:(3)$$\begin{array}{c} \Sigma_{n}^{2}\left(x\right)=\langle \;{𝜩}_{n}\left(x\right)\;{𝜩}_{n}^{T}\left(x\right)\;\rangle ={𝖫}_{n}\left(x\right)\;{𝖫}_{n}^{T}\left(x\right)\;\;\;E_{n}^{2}\left(x\right)=\;\;\mathrm{Tr}\;\left(\Sigma_{n}^{2}\left(x\right)\right)=\;\;\mathrm{Tr}\;\left({𝖫}_{n}^{T}\left(x\right){𝖫}_{n}\left(x\right)\right) \end{array}$$
The matrix ${𝖫}_{n}\;{𝖫}_{n}^{T}$ has the same eigenvalues as the Lyapunov matrix ${𝖫}_{n}^{T}{𝖫}_{n}$ but their eigenvectors are different. The invariants $I_{n}^{\left(k\right)}\left(x\right)$ of the Lyapunov matrix are equal to the sum of the dk products of the eigenvalues and can be computed with the Faddeev–Leverrier algorithm. From a geometrical viewpoint, the invariants are equal to the sum of the squared volumes of the dk parallelotopes whose edges are ${𝖫}_{n}\left(x\right)e_{j}$, and $e_{j}$ are any orthonormal base vectors. According to the Oseledec theorem, by writing the Lyapunov matrix as ${𝖫}_{n}^{T}\left(x\right){𝖫}_{n}\left(x\right)={𝖶}_{n}\left(x\right)\;e^{2n\;\Lambda_{n}\left(x\right)}\;{𝖶}_{n}\left(x\right)$, the diagonal matrix $\Lambda_{n}\left(x\right)$ and the eigenvectors matrix ${𝖶}_{n}\left(x\right)$ have a limit, for n→∞ is equal to Λ(x) and 𝖶(x), respectively. As a consequence, letting $\Lambda \left(x\right)=\;\;\mathrm{diag}\;\left(\lambda_{1}\left(x\right),\ldots ,\lambda_{2d}\left(x\right)\right)$, we have
(4)$$\begin{array}{c} \lim_{n\to \infty }\;\frac{1}{2n}\;\log I_{n}^{\left(k\right)}\left(x\right)=\lambda_{1}\left(x\right)+\lambda_{2}\left(x\right)+\ldots +\lambda_{k}\left(x\right) \end{array}$$
If the map *M* is symplectic, the exponents are pairwise opposite $\lambda_{d+j}=-\lambda_{d-j+1}$. If the map is integrable, then the first *d* eigenvalues of the Lyapunov matrix grow as $n^{2}$, while the last *d* decrease as $n^{-2}$, so that all the Lyapunov exponents vanish. For a generic map, the power law growth or exponential growth with *n* of the first *d* invariants guarantee the classification of the phase space regions of regular and chaotic evolution. Below, we briefly introduce the reversibility error. For a complete description of the indicators and their properties, see Appendix B.

### Reversibility Error

We consider the Backward–Forward process (BF) as one implementation of the Reversibility error indicator. In this case, we iterate *n* times the map with a random perturbation of amplitude ϵ first, and then the inverse map. The recurrence is given by
(5)$$\begin{array}{c} y_{n^{\prime }}=M\left(y_{n^{\prime }-1}\right)+\epsilon \xi_{n^{\prime }}\;1\leq n^{\prime }\leq n\;\;y_{n^{\prime }}=M^{-1}\left(y_{n^{\prime }-1}\right)\;n+1\leq n^{\prime }\leq 2n \end{array}$$
and the linear response is defined by
(6)$$\begin{array}{c} {𝜩}_{2n}^{BF}\left(x\right)=\lim_{\epsilon \to 0}\frac{y_{2n}-x}{\epsilon }=\sum_{k=1}^{n}\;{𝖫}_{k}^{-1}\left(x\right)\;\xi_{k} \end{array}$$
where ${𝖫}_{k}^{-1}$ is the inverse of the matrix ${𝖫}_{k}$ defined by (Equation 1). In the absence of perturbation, we are back to the initial condition $y_{2n}\equiv x_{2n}=x$. The covariance matrix of the random vector ${𝜩}_{2n}^{BF}$ is given by
(7)$$\begin{array}{c} \Sigma_{BF\;n}^{2}\left(x\right)=\langle {𝜩}_{2n}^{BF}\left(x\right)\;{\left({𝜩}_{2n}^{BF}\right)}^{T}\rangle =\sum_{n^{\prime }=1}^{n}\;{\left({𝖫}_{n^{\prime }}^{T}\left(x\right){𝖫}_{n^{\prime }}\left(x\right)\right)}^{-1}\; \end{array}$$
The square of the BF reversibility error $E_{BF\;n}$ is defined as the trace of $\Sigma_{BF\;n}^{2}$. The asymptotic limit of the invariants $I_{BF\;\;n}^{\left(k\right)}\left(x\right)$ is the same as (Equation 4) for k≤d. The log of the last invariant $I_{BF\;\;n}^{\left(2d\right)}\left(x\right)$ is the Gibbs entropy of the process with the covariance matrix $\Sigma_{BF\;n}^{2}\left(x\right)$ and its asymptotic behavior is the same as the Kolmogorov–Sinai entropy.

If the map is symplectic, then the matrices ${𝖫}_{n^{\prime }}{𝖫}_{n^{\prime }}^{T}$ and ${𝖫}_{n^{\prime }}^{T}{𝖫}_{n^{\prime }}$ are symplectic. Therefore, the trace of ${𝖫}_{n^{\prime }}^{T}{𝖫}_{n^{\prime }}$ and its inverse are equal. In this case, the BF reversibility error is simply related to the Lyapunov error $E_{n}$ as
(8)$$\begin{array}{c} E_{BF\;\;n}^{2}\left(x\right)=\sum_{n^{\prime }=1}^{n}\;E_{n^{\prime }}^{2}\left(x\right) \end{array}$$
Previously, the BF process has been considered with the noise applied to both the B and F iterations. The covariance matrix of the linear response in in this case is given by
(9)$$\begin{array}{c} \Sigma_{BF\;\;n}^{2}\left(x\right)=\frac{1}{2}\langle {𝜩}_{2n}^{BF}\left(x\right)\;\left({𝜩}_{2n}^{BF}\right)\rangle =\frac{1}{2}\;\;I+\sum_{n^{\prime }=1}^{n-1}\;{\left({𝖫}_{n^{\prime }}^{T}\left(x\right){𝖫}_{n^{\prime }}\left(x\right)\right)}^{-1}+\frac{1}{2}\;\;{\left({𝖫}_{n}^{T}\left(x\right){𝖫}_{n}\left(x\right)\right)}^{-1} \end{array}$$
and proof of (Equation 9) can be found in [3] Section 2.3. The asymptotic behavior of the invariants of the reversibility error covariance matrix $\Sigma_{nBF}^{2}$ is determined by the positive Lyapunov exponents. Indeed, the limit of ${\left(2t\right)}^{-1}\log\left({𝖨}_{nBF}^{\left(k\right)}\right)$ for t→∞ is the sum of the positive exponents among the first *k*. As a consequence, $\frac{1}{2}\log\left(I_{nBF}^{\left(2d\right)}\right)$, which corresponds to the Gibbs entropy of the BF random process where $I_{nBF}^{\left(2d\right)}=det(\Sigma_{nBF}^{2})$, is the sum of all the positive Lyapunov exponents (the first *d* for a symplectic map), just as the Kolmogorov–Sinai entropy. Notice that the difference with the previous definition and (Equation 7) is negligible for a large *n*. Such a definition was initially proposed to compare the reversibility error due to a small random displacement with respect to the reversibility error due to the round off. The Reversibility Error indicator due to round off, denoted by REM, is a numerical implementation of $E_{n}^{R}\left(x\right)$ and numerical equivalence has been proven for simple maps [34].

Denoting with $M_{\epsilon }$ the map evaluated with round off and $M_{\epsilon }^{-1}$ the inverse map evaluated with the round off, the REM indicator is then
(10)$$\begin{array}{c} REM_{BF\;\;n}\left(x\right)=\frac{∥M_{\epsilon }^{\circ \;\;-n}\circ M_{\epsilon }^{\circ \;\;n}\left(x\right)-x∥}{\sqrt{2}\;\;\epsilon } \end{array}$$
The round off is a pseudo-random process in which just one realization is available. When different values for *n* or x are used, it is evident that $REM_{BF\;\;n}$ exhibits significant fluctuations with respect to $E_{BF\;\;n}$, which is the result of an averaging process over the random displacements. No higher invariants can be defined for the reversibility error due to round off.

We can also consider the specular implementation, namely the Forward–Backward $REM_{FB}$, which consists of iterating *n* times first the perturbed inverse map $M^{-1}$ and then the map *M*. For an autonomous symplectic map however, all the invariants are the same as for the BF process. Moreover, for a Hamiltonian flow, equivalence of the FB and BF invariants is a consequence of the time reversal invariance.

The linear approximation given by $y_{n}=x_{n}+\epsilon {𝜩}_{n}$ can be considered; however, since the remainder of order $\epsilon^{2}$ can generically be neglected only for a number of iterations, small with respect to log(1/ϵ), it is of no practical use. The linear response being based on the ϵ→0 limit is valid for any number *n* of iterations.

## 3. Numerical Results for the 2D Billiard

In this section, we introduce numerical results for a 2D billiard; the reflection map for a convex billiard defined by (Equation 22) with $f\left(x\right)=x^{2}/3$ is presented. The billiard analyzed in this section has a closed boundary for $\epsilon \leq 1/\sqrt{3}∼0.577$ as one can easily show. Indeed, letting $V\left(x\right)=x^{2}/2+\epsilon x^{3}/3$, the maximum occurs for x=−1/ϵ and we require $V\left(-1/\epsilon \right)=1/\left(6\epsilon^{2}\right)\geq 1/2$ to have a closed boundary so that $\epsilon \leq 1/\sqrt{3}$. We have restricted our analysis to ϵ≤/1/2, observing that for ϵ≤0.1, the map is almost integrable; for ϵ=0.2, the area of chaotic regions is significant and already for ϵ=0.4, a large fraction of the phase space is chaotic. We have used the coordinates (ϕ,p) in the phase space even though the map preserves the measure but is not area preserving. We have compared the phase portraits with the Lyapunov error $E_{n}\left(x\right)$, the reversibility error $E_{n}^{BF}$ and the round-off induced reversibility error $REM_{n}^{BF}$, computed according to (Equation 38), (Equation 48) and (Equation 51), respectively. In Figure 1, we show the billiard for:

We have analyzed the orbits for the billiard, comparing them with the color map of the Lyapunov error $E_{n}\left(x\right)$, reversibility error $E_{n}^{BF}\left(x\right)$ and the round-off induced reversibility error $REM_{n}^{BF}\left(x\right)$, where $x={\left(\phi ,p\right)}^{T}$ is chosen on a regular grid of the $N_{g}\times N_{g}$ points. A logarithmic color map is used to show the results. By defining the tangent vector τ(s)=∂r(s)/∂s and the ray velocity as ∥v(s)∥, then the range of the phase ϕ of the position vector is [0,2π], and the range of the momentum p=τ·v is [−1,1]. Since the orbits are symmetric, by changing *p* into −p, the chosen range for *p* is then in the range [0,1]. For a deeper discussion on the properties of the billiard, see Appendix A.

It is important to notice that the indicators give additional information with regard to the phase portrait, since they provide a quantitative measurement of the chaos for an orbit. Additionally, for 4D billiards Poincare’ sections are not available and the indicators are the only methods to investigate the stability of the phase space.

In Figure 2, we compare the phase portrait against the $E_{n}\left(x\right)$ for the billiard with ϵ=0.1. The correspondence between the phase portrait and $E_{n}\left(x\right)$ color plot is good and in both cases, a thin layer of chaotic orbits is seen at the boundary of the main chains of islands. In the interior of the chains of islands, the error tends to be zero because the de-tuning (derivative of the frequency with respect to the action) is low. By approaching the separatrix, the de-tuning grows, diverging on the separatrix itself, where the error growth with *n*, changing from a power law to an exponential one.

In Figure 3, $E_{n}^{R}\left(x\right)$ and $REM_{n}^{BF}\left(x\right)$ show a similar behavior. REM is similar to the error induced by the small random displacements, but exhibits higher fluctuations because the averaging over the random process is missing. It is worth noticing that ${\left(E_{n}^{R}\right)}^{2}$ constitutes the sum along the orbit of ${\left(E_{n}\right)}^{2}$, provided (s,p) are used as canonical coordinates. The Lyaponov error oscillates with *n* in the regions of quasi-integrable motion and for fixed *n*, oscillations are observed in phase space. These oscillations can be eliminated by using the MEGNO average (see, for instance, Ref. [35]) and they disappear for $E_{n}^{R}\left(x\right)$.

In Figure 4 and Figure 5, the Poincare section, $E_{n}^{R}\left(x\right)$, $E_{n}\left(x\right)$ and $REM_{n}^{BF}\left(x\right)$ are shown for a larger perturbation ϵ=0.2. The larger deformation with respect to the integrable billiard increases the area of the chaotic orbits. For ϵ=0.3, almost one half of the unit phase space area—in the ($\phi_{0}/2pi$, $p_{0}$) initial coordinates used in the figures—is filled with chaotic orbits and this fraction increases approaching 1, when ϵ tends to the limit value $\epsilon =1/\sqrt{3}$. For a 2D map, the phase space portraits provide the required information on the orbits stability; however, one advantage of the proposed indicators is that they provide a quantitative value to discriminate between regular and chaotic orbits. To explore small details in the phase space, one can analyze a smaller region and increase the number of iterations. Moreover, the indicators become the unique stability analysis tool when the dimensionality of the problem is increased. This is the case for the 3D billiard which leads to a 4D reflection map. In this case, the 2D phase portraits are no longer available.

## 4. Conclusions

We have considered the motion of particles on a 2D billiard with a convex reflecting boundary using a parametrization different from the one proposed by [36]. Our implementation can be easily extended to a 3D billiard. The computation of the arc length *s* on the boundary requires a numerical integration, which can be avoided choosing the phase ϕ of the position vector r rather then the curvilinear abscissa *s*, even though, in this case, the map is only measure preserving. The stability of the orbits has been analyzed using the Lyapunov and Reversibility error fast indicators. Both indicators are invariant with respect to the choice of the initial deviation and of the orthogonal reference frame. We have shown that the logarithm of the second reversibility error invariant is the Gibbs entropy of the random vector defining the deviation from reversibility. The numerical results of the Lyapunov and reversibility error indicators, presented for a selected family of billiards, confirm the reliability of the proposed methods to explore the sensitivity of ray propagation to small random perturbations.

## Figures and Tables

**Figure 1 entropy-25-01251-f001:**
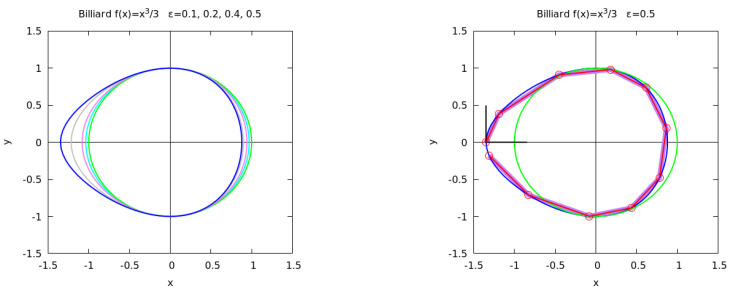
**Left panel**: billiard boundary defined by (Equation 22) with $f\left(x\right)=x^{2}/3$ and ϵ = 0, 0.1, 0.2, 0.4, 0.5 (green, cyan, purple, grey, blue). **Right panel**: rays trajectory for ϵ=0.5 and n=10 (purple line) and the reversed ray trajectory (red line). At the initial point, the tangent and inner normal vectors are shown (black lines).

**Figure 2 entropy-25-01251-f002:**
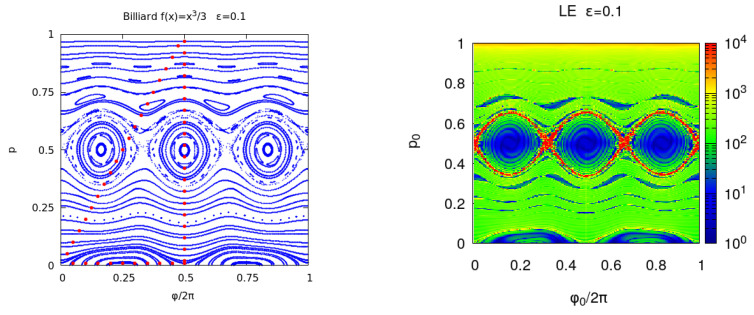
**Left panel**: phase portrait of the billiard with ϵ=0.1. Each orbit is computed for n=1000 and the initial points of each orbit are the red dots. **Right panel**: Lyapunov error $E_{n}\left(x\right)$ where $\phi_{0}/\left(2\pi \right)$ and *p* are chosen in a regular grid of the unit square with $N_{g}\times N_{g}$ points and $N_{g}=200$. The iterations number is n=200 and the results are shown in a logarithmic color scale.

**Figure 3 entropy-25-01251-f003:**
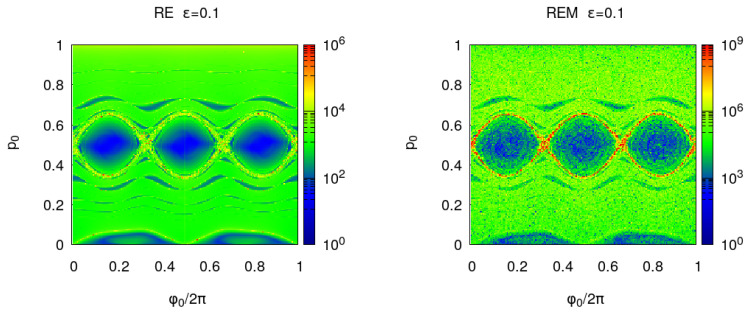
**Left panel**: billiard with ϵ=0.1 reversibility BF error $E_{n}^{BF}\left(x\right)$ where $\phi_{0}/\left(2\pi \right)$ and *p* are chosen in a regular grid of the unit square with $N_{g}\times N_{g}$ points and $N_{g}=200$ and the iterations number is n=200. **Right panel**: round-off induced BF reversibility error $REM_{n}^{BF}\left(x\right)$.

**Figure 4 entropy-25-01251-f004:**
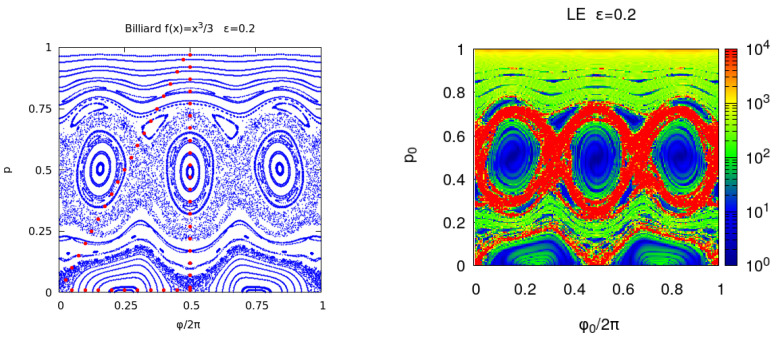
**Left panel**: phase portrait of the billiard with ϵ=0.2. Each orbit is computed for n=1000 and the initial points of each orbit are the red dots. **Right panel**: Lyapunov error $E_{n}\left(x\right)$ where $\phi_{0}/\left(2\pi \right)$ and *p* are chosen in a regular grid of the unit square with $N_{g}\times N_{g}$ points and $N_{g}=200$. The iterations number is n=200 and the results are shown in a logarithmic color scale.

**Figure 5 entropy-25-01251-f005:**
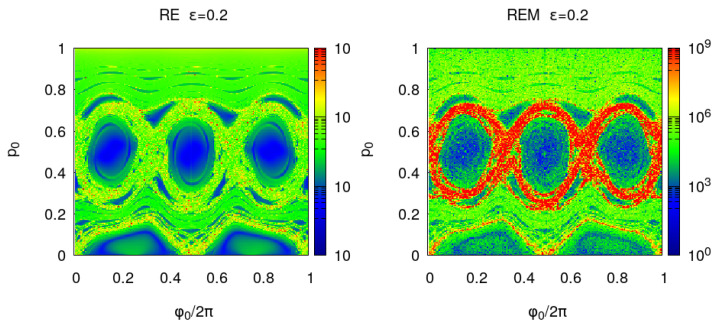
**Left panel**: billiard with ϵ=0.2 reversibility BF error $E_{n}^{BF}\left(x\right)$ where $\phi_{0}/\left(2\pi \right)$ and *p* are chosen in a regular grid of the unit square with $N_{g}\times N_{g}$ points and $N_{g}=200$ and the iterations number is n=200. **Right panel**: round-off induced reversibility error $REM_{n}^{BF}\left(x\right)$.

## Data Availability

Not applicable.

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
