# Peer review of "Chaos Detection by Fast Dynamic Indicators in Reflecting Billiards"

_entropy, 2023, doi:10.3390/e25091251_

Round 1

Reviewer 1 Report

Report for the Manuscript
Energy ray transport and chaos detection by dynamic indicators in absorbing billiards

by Gabriele Gradoni, Giorgio Turchetti and Federico Panichi

The authors of the manuscript "Energy ray transport and chaos detection by dynamic indicators in absorbing billiards" provide a detailed presentation of Lyapunov error, the reversibility errors, reflection maps and others for nearly circluar maps in 2D and 3D and illustrate it in the 2D case by numerical evaluation.

The paper starts with a introduction. Section 2 derives in great detail (~7 pages) various presentations of (linearized) maps for forward and backward propagation on billiards derived from the circular geometry in 2D and 3D. The following section 3 uses this results to derive Lyapunov and reversibility error indicators. These results are applied by numerical evaluation for the 2D case in the following section. Section 4 demonstrates that all measures considered in this manuscript agree well with the phase portrait of the system. Section 5 considers transport from a localized source for two examples, namely the circle and the Robnik billiard, both with and without absorption.

The manuscript (re)derives methods and strategies used in the description of regular and chaotic systems such as Lyapunov exponents and other measures. However, my impression is that the paper stops at the most interesting physical points after going through all the maths in detail (though sometimes in a special notation). I recommend to refine the manuscript by extending the physical interpretation of the results so that the title of the paper is justified. Right now the presentation of the physical results in Figs. 7 and 8 is done within only 6 lines.

Here are some questions:

1) The emission from the source shown in Figs. 7 and 8 contradicts my intuition. Even if the source is placed right at the boundary, I would expect a (half) circular charactersitics of a pointlike isotropic source that should be visible in the ray intensity. However, it is not. Why?
Why are the whispering gallery trajectories so pronounced? What happens to the other rays emerghing from the source?

2) Related to this question is the question concerning the impact of an absorbing material. The source characteristics seems to be more "typical" then - what is the reason for this? What is the origin for the other structures visible (see Figs. 7 and 8, right panels) and how is it affected by having an int integrable vs. a chaotic geometry?

3) What can be said about transport, i.e. about the temporal evolution of the results shown in Figs. 7 and 8? How does it depend on T?

4) The authors state in the paragraph after Eq. 71:
"The Lyaponov error can provide relevant information on the relation between the sensitivity of the initial conditions and the time variation of the energy density. Chaos in the reflections dynamics can influence the illumination pattern."
This is an interesting result and deserves to be explained in more detail and ideally to be illustrated in a figure.

5) The authors state the advantage of using the variable phi rather than the arc length in their considerations. This is an important statement (as it deviates from the typical Birkhoff convention) and should be explained in more detail.

Additional changes required:
Read start of Section 4 carefully, there are confusing typos/errors. Please check for further typos in formulae.

Changes to be considered:
Generalization from the 2D to the 3D case is done in Section 2, but not used later on in the manuscript. I suggest to omit the 3D case or put it into an Appendix.

Typos spotted:
p. 5 unlikely should propably read unlike
p. 11 in Eq. (50) the linear response should probably always have the argument
p. 12 right after start of section 4: inconsistencies
p. 15: Fig. 6 "eft panel"
p. 16: an it is convenient --> and it is convenient
p. 20: Monte-Carlo _what_

Author Response

Comment: The manuscript (re)derives methods and strategies used in the description of regular and chaotic systems such as Lyapunov exponents and other measures. However, my impression is that the paper stops at the most interesting physical points after going through all the maths in detail (though sometimes in a special notation). I recommend to refine the manuscript by extending the physical interpretation of the results so that the title of the paper is justified. Right now the presentation of the physical results in Figs. 7 and 8 is done within only 6 lines.

Answer: We refined the title and the body of the paper so to acknowledge the reviewer’s feedback. One of the main purposes of the paper is to be an implementation of the fast indicators for an interesting physical problem, i.e., the billiard and ray propagation. Concerning the energy transport by the rays emitted from a source, we outlined how to compute it in the vacuum or a uniform absorbing medium. The results presented in figures 7 and 8 are preliminary. An obvious effect of the absorption is the decay of the illumination with the distance from the source and the appearance of some patterns which are not observed in the vacuum. A deeper investigation is needed for their physical interpretation.

Question: The emission from the source shown in Figs. 7 and 8 contradicts my intuition. Even if the source is placed right at the boundary, I would expect a (half) circular characteristics of a point-like isotropic source that should be visible in the ray intensity. However, it is not. Why? Why are the whispering gallery trajectories so pronounced? What happens to the other rays emerging from the source? Related to this question is the question concerning the impact of an absorbing material. The source characteristics seems to be more ”typical” then - what is the reason for this? What is the origin for the other structures visible (see Figs. 7 and 8, right panels) and how is it affected by having an int integrable vs. a chaotic geometry?

Answer: The time evolution of the illumination pattern might be visualized with a sequence of images like figures 7 and 8 for a sequence of values of T. However this is post-posed to a future work due to the limited time for the revision process. Question: What can be said about transport, i.e. about the temporal evolution of the results shown in Figs. 7 and 8? How does it depend on T?

Answer: due to the interesting detail of the transport, its application to complex physical problems and the lack of time to expand on the mathematics, we decided to omit this section and describe it in a separate, dedicated paper.

Question: The authors state in the paragraph after Eq. 71: ”The Lyaponov error can provide relevant information on the relation between the sensitivity of the initial conditions and the time variation of the energy density. Chaos in the reflections dynamics can influence the illumination pattern.” This is an interesting result and deserves to be explained in more detail and ideally to be illustrated in a figure.

Answer: the theoretical framework to investigate the relation between the sensitivity of the orbits to chaos and the time variation of the energy density has been established. However, to answer to this question in a satisfactory way requires a dedicated project. This is beyond the scope of this paper and requires more than a single section or a new figure. We decided to dedicate to this relation a new stand-alone project.

Question: The authors state the advantage of using the variable phi rather than the arc length in their considerations. This is an important statement (as it deviates from the typical Birkhoff convention) and should be explained in more detail.

Answer: We have acknowledged that this parametrization was introduced decades ago in ”B. Dietz, U. Smilansky, Chaos 3, 581 (19930)”. As pointed out there, the parametrization is convenient when dealing with semi-classical and ray based formulations in specific (convex) geometries. This part has now been moved to an Appendix.

Additional changes required: Read start of Section 4 carefully, there are confusing typos/errors. Please check for further typos in formulae.

Answer: we made substantial rephrasing and re-organization of the paper, including two appendices for the mathematical formulation of the problem and for the detailed description of the indicators.

Changes to be considered: Generalization from the 2D to the 3D case is done in Section 2, but not used later on in the manuscript. I suggest to omit the 3D case or put it into an Appendix.

Answer: we acknowledged the comment of the reviewer and we omit the 3D case.

Reviewer 2 Report

Report on:
"Energy ray transport and chaos detection by dynamic indicators in absorbing billiards"
 by Gabriele Gradoni, Giorgio Turchetti and Federico Panichi (05/2023)
----------------------------------------------------------------------

The present effort concerns the propagation of rays in a billiard which, in the eikonal approximation, models the propagation of electromagnetic waves in a closed domain with a convex reflecting boundary.The stability of the orbits has been investigated by recourse to the Lyapunov and reversibility error invariant indicators, which provide an estimate of the sensitivity to both a small initial random deviation and a small random deviation at any reflection.

The proposed implementation can be easily extended to a 3D billiard, where two Lyapunov and reversibility error invariant indicators -their asymptotic behaviour depending on the first two non negative Lyapunov exponents- are coonsidered. Further, their efficiency to detect chaos has been tested in the case of a family of chaotic billiards.

In the realm of the transport problem the rays and particles in a uniform medium have been considered, which have the same path and time law in the case of free propagation within the billiard. The authors propose a numerical procedure to compute the ray energy density or particles density even when a uniform absorption takes place.

The manuscript is well written and properly organized; in fact, the manuscript structure flows. The grounds and objetives are clearly stated. The outcomes of numerical experiments are displayed in neat plots and satisfactorily explained. The interpretation of the results and the study conclusions are sound, and the strengths of the investigation are clearly emphasized. The references are relevant and cover both well-known historical literature and more recent developments.

Therefrom, I deem the manuscript adequate for its publication in Entropy in its present form.

"""""""""""""""""""""""""""""""""""""""""""""""""""""""""""""""""""""""""""""""""""""""""""""""""""""""""""""

Some minor corrections:

l. 58:        iteration   ->   iterations

l. 76:       , an    ->   , and an

second line after eq. (1):        Denting   -> Denoting

after eq. (2):            tangent   ->   tangents

in eq. (5):       phi   ->  \phi

below eq: (18):             13   ->   (13)

line upon eq. (31):            27   ->   (27)

fist line below eq. (33):        32  ->  (32)

third line in Section 4:       $x = -1\epsilon$     ->   $x = -1/\epsilon$

last line in first paragraph in Section 4:   for... ???
                         please finish the sentence

l. 158:        filled    ->   is filled

last line in p.14:             equation equation ...

first line in p. 17:           $(x_t, y_t, \theta_t)$
                it says y_y
second line below eq. (65):        then then

fifth line after (66):       uniforms   ->   uniform

first line after (67):       vary   ->   very

in line before eq. (69):       time $t$   ->   time $t$ are

third line below (70):        its  ->  it is

last line in previous to last paragraph in p. 18:    that  ->   than

first line below (71):        an   ->   and

fourth line below (71):       Lyaponov   ->    Lyapunov

line 218:       then    ->   than         

Author Response

Answer: we acknowledge the minor corrections provided by the reviewer.

Reviewer 3 Report

In the manuscript the authors first review convex 2D billiards and two different parametrizations, then Lyapunov and reversibility indicators, and then ray transport. I do not recommend publication of the manuscript in its present form.

1) The title has not much to do with the content. The authors mention for the first time absorbing media on page 18 in the last paragraph before the Conclusion, and just show some results.

2) The authors claim that they introduce a new parametrization. That's not correct. It is a commonly used parametrization. I learned about it decades ago in a paper from 1993 (B. Dietz, U. Smilansky, Chaos 3, 581 (19930), but maybe there are even earlier works.

3) Section 2 doesn't contain any new results. These are well known procedures. They should be removed or moved to the appendix, also because the authors start to introduce less known procedures in Sect. 3 on page 9 only. Results for 3D billiards are not shown, and above all, the content of Sect. 2.3 is well-known. It should be removed.

4) Section 3 is not well written. The authors should concisely define the indicators that they use. Are the eigenvalues lambda_i the Lyapunov exponents, and is the Lyapunov error given by the sum of the Lyapunov exponents? That is, isn't the 'Lyapunov error' directly related to the Kolmogorov-Sinai entropy (V. Latora and M. Baranger, Phys. Rev. Lett. 82, 520 (1998)).

5) The authors should explain in more detail and concisely this indicator and also the other ones and outline -- some steps in their 'derivations' are either superfluous or should be moved to the appendix, others like starting from Eq. (47) are too short and incomprehensible.

6) It is not clear to me what exactly is new in the manuscript. In Figures 4 and 5 the Poincare surface of section is compared to the error indicators. If the authors want to demonstrate that the indicators do better than the PSOS, then it is not convincingly explained.

7) What is the intention of Section 5? It comes from nowhere. It needs to be revised. Also, suddenly, the authors mention mass-less relativistic particles without appropriately introducing changes needed in the definition of the associated dynamics, transformation, parametrization, etc.. They better remove parts related to the relativistic case, because as it stands it is wrong, and the topics of the manuscript are nonrelativistic billiards. Even for the quantum limit confinement of mass-less relativistic particles to a billiard would need further explanation (Berry and Mondragon, 1987).

8) Generally Sect. 5 is unreadable because the quantities of interest are not well introduced, also Figs. 7 and 8, are shown without further explanation. However, this section and Sect. 3 seem to be the relevant section of the manuscript.

- The manucsript contains typos.
Furthermore the notation '\vec v(s+0)' is confusing.
In Eq. (2) it should be '=' instead of '\equiv ='. Eq. (5) 'phi'.
Parts of Figure 1 look weird. Does 't_\ast' after Eq. (20) denote the solutions of Eq. (20)? Is \vec v_n^2=1?
Line 124 ‘-1\epsilon’. Line 129-131 are repeated in line 132-…. Notation of boldface ‘\Phi’ should be changed to ‘\Phi’ as it is not a vector.
Also above Eq. (57) amplitude of \vec c(\vec r) should not be boldface.

 - The list of references is incomplete.
It is by far not sufficient and maybe even inappropriate to mention in connection with microwave billiards only the book by Stoeckmann.
Experiments have been performed in the group of Sridhar (PRL 67, 785 (1991), of Stoeckmann in Marburg, of Richter in Darmstadt, of Sirko in Warsaw, of Anlage at University of Maryland, of Kuhl in Nice,...).
Reference 12 should be replaced by the original works by Sinai and Bunimovich.
Reference 28 is incomplete. I suppose that the authors mean Marko Robnik, not Robnic.

see report. I did not list all typos.

Author Response

Please find detailed answers attached.

Round 2

Reviewer 3 Report

The authors replied appropriately to the points of criticism and suggestions, and the readability and, generally, style of the manuscript have improved considerably. I recommend publication after the following minor corrections have been incorporated:

The authors should check carefully all formulas:
- In a few cases they use M^n, more often they use M^{\circ n}
- In a few cases BF appears as subscript, however, dominantly as superscript to \Xi
- In most cases \Sigma^2, is defined as the average of the product of \Xi and \Xi^T, in some cases as the product of \Xi with itself.
- In some cases 'L' has no index.
- In line 47 E_n and also E_n^R depend on x. The arguments are dropped in some places
- In equation (A42) it should be '2n' in the argument of the exponential and the index of W^T is missing.
- Above Eq. (8) / below Eq. (A51): 'If the map is symplectic, ??? and L^T_n^\primeL_\n^\prime are symplectic matrices.'. Do the authors mean $L_n^\prime L^T_\n^\prime$?
- Below Eq. (A42) it should be $\lambda_{d-j+1}(n)$.
- Above Eq. (A55): the '=' in front of 'is replaced' should be deleted. I do not understand the derivation following then. Generally, the authors should either change or explain why it's different, otherwise it's confusing.
- Furthermore, the sentence starting with 'For a 3D...' in the abstract, line 6 should be deleted, because 3D biliards are not treated and, above all, this is known.
- The statement in the first sentence of the introduction has been demonstrated already in 'E. Doron et al., PRL 65, 3072 (1990) for two-dimensional open elecromagnetic cavities and in C. Dembowski et al., PRL 89, 064101 (1989) for closed three-dimensional cavities. Therefore, the authors should refer to [1] as a review on that topic. Furthermore, the semi-classical treatment of electromagnetic cavities has been done by Balian with Duplantier and others in the 70s.
- Line 19: it should be 'time-reversal invariance violation at and around exceptional points'
- Line 24: What is the abbreviation 'EMC' for?
- Line 95: It should $x= -1 / \epsilon$
- Line 108: Tau and v are not defined. (only in the appendix)
- Line 124: I don't understand the sentence starting with 'It is also worth...'
- I strongly suggest to remove appendix A.3, because 3D billiards are not treated, so that nobody will read that part, and the derivation has been done before and is straightforward with the procedure for the 2D case.

The manuscript contains some grammar errors and typos

Author Response

We are deeply thankful to the Reviewers for the detailed comments that have helped us to substantially imporve the quality of the manuscript. 

We have addressed all the comments and made each and any minor amendments to the manuscript as suggested. 

Answers to more substantial points can be found in the attached pdf. 
